

# Exotic lagomorph may influence eagle abundances and breeding spatial aggregations: a field study and meta-analysis on the nearest neighbor distance

Facundo Barbar[1], Gonzalo O. Ignazi[1], Fernando Hiraldo[2] and Sergio A. Lambertucci[1]

[1] Grupo de Biología de la Conservación, Ecotono Laboratory, INIBIOMA—CONICET (Universidad Nacional del Comahue), San Carlos de Bariloche, Río Negro, Argentina
[2] Departamento de Biología de la Conservación, Estación Biológica Doñana-CSIC España, Sevilla, España

## ABSTRACT

The introduction of alien species could be changing food source composition, ultimately restructuring demography and spatial distribution of native communities. In Argentine Patagonia, the exotic European hare has one of the highest numbers recorded worldwide and is now a widely consumed prey for many predators. We examine the potential relationship between abundance of this relatively new prey and the abundance and breeding spacing of one of its main consumers, the Black-chested Buzzard-Eagle (*Geranoaetus melanoleucus*). First we analyze the abundance of individuals of a raptor guild in relation to hare abundance through a correspondence analysis. We then estimated the Nearest Neighbor Distance (NND) of the Black-chested Buzzard-eagle abundances in the two areas with high hare abundances. Finally, we performed a meta-regression between the NND and the body masses of Accipitridae raptors, to evaluate if Black-chested Buzzard-eagle NND deviates from the expected according to their mass. We found that eagle abundance was highly associated with hare abundance, more than with any other raptor species in the study area. Their NND deviates from the value expected, which was significantly lower than expected for a raptor species of this size in two areas with high hare abundance. Our results support the hypothesis that high local abundance of prey leads to a reduction of the breeding spacing of its main predator, which could potentially alter other interspecific interactions, and thus the entire community.

## INTRODUCTION

The spatial distribution of a species is determined by extrinsic and intrinsic factors. Resource availability is the main extrinsic factor that may influence spatial distribution of organisms (*Guisan & Zimmermann, 2000*; *Guisan & Thuiller, 2005*). Changes in food sources could be modifying consumers' spatial distribution. Ecosystems are composed of different species that consume resources that are naturally limited (*Chase & Leibold, 2003*).

Corresponding author
Facundo Barbar,
facundo.barbar@gmail.com

Within a given trophic level, interspecific and intraspecific interactions emerge in order to use these resources. These include agonistic interactions such as direct competition, spatial exclusion and intraguild predation (*Amarasekare, 2003*; *Sergio & Hiraldo, 2008*), as well as resource partitioning that favors species coexistence (*Martin, 1996*; *McDonald, 2002*; *Griffin et al., 2008*). At the individual level, the exclusion of conspecifics leads to territoriality, eventually reaching a spatial configuration that maximizes the number of territories in a given area as a function of resource availability (*MacLean & Seastedt, 1979*; *Schoener, 1983*).

One of the main intrinsic factors limiting the spatial distribution of species is animal body mass, as larger species require more energy to fulfill their energetic metabolic requirements (*Damuth, 1981*; *Peters, 1986*; *White et al., 2007*). In any guild (e.g., carnivores, raptors), the difference in body mass of the various species is the main factor driving resource partitioning (*Aljetlawi, Sparrevik & Leonardsson, 2004*; *Brose, 2010*), as consumers select prey that provide a positive energetic balance between food intake and handling time (*Brose et al., 2006*; *Allhoff & Drossel, 2016*). This process of prey selection is directly linked to competing species coexistence (*Loreau & Hector, 2001*; *Amarasekare, 2002*). On the other hand, this energetic constraint also implies that larger species may require larger territories to provide enough resources, therefore spacing their territories more widely than those of smaller species (*Schoener, 1968*; *Peery, 2000*).

In the current global change scenario, humans are responsible for altering the ecosystems in several ways and these changes are occurring in an accelerated way (*Barnosky et al., 2012*). The introduction of species is among the main factors of global change, which is not only homogenizing biodiversity at a global scale but also has the potential of altering energy fluxes (*McKinney & Lockwood, 1999*; *Newsome et al., 2015*). The introduction of exotic species may profoundly impact the relative abundance of native species and therefore community structure (*Vitousek, 1990*; *Vitousek et al., 1997*; *Tilman, 1999*; *Newsome et al., 2015*), which may favor some native species over others, improving their population parameters. However, this change in structure can lead to unbalanced ecological situations (e.g., *Tablado et al., 2010*; *Speziale & Lambertucci, 2013*).

Patagonia, as the southern tip of South America, is one such region to have suffered multiple species introductions (*Rodríguez, 2001*). One of the most conspicuous invaders has been the European hare (*Lepus europaeus*) which reached the region in the early 1900s (*Grigera & Rapoport, 1983*). European hares had no other similar species in the region and became extremely abundant in number over a short period of time (*Bonino, Cossíos & Menegheti, 2010*). As such, this introduced species may potentially alter energy fluxes, trophic interactions and indirectly change community structure (*Simberloff & Von Holle, 1999*; *Simberloff et al., 2013*). In fact, there is evidence that many predators in Patagonia have already shifted their diets to include this new and abundant source of food (*Monserrat, Funes & Novaro, 2005*; *Zanón Martínez et al., 2012*; *Barbar, Hiraldo & Lambertucci, 2016*).

Top predators that depend upon scarce resources are adequate to explore the resource availability-territory size relationship, as their territories cover greater areas than herbivorous species (*Schoener, 1968*) and any change can be easily quantified with simple metrics such as the Nearest Neighbor Distance (NND, *Clark & Evans, 1954*). This

includes raptor species that generally behave as central place foragers and whose territory sizes are determined by resource abundance (*Sonerud, 1992*; *Newton, 2010*). Their fidelity to nesting areas means that the geographical distance between breeding sites can be used to quantify the relationship between resource availability and territory size and location.

Here we aim to explore how the increased abundance of an exotic species (the European hare) may influence the raptor guild at the higher tropic level, paying particular attention to the Black-chested Buzzard-eagle (*Geranoaetus melanoleucus*; hereafter BCB eagle), which is the species that consumes it the most (*Barbar, Hiraldo & Lambertucci, 2016*). For this we first quantified and compared the abundance of different raptor species to the abundance of hares in Northwestern Patagonia. We then determined the NND for the BCB eagle in an area of high exotic hare population density. Finally, we compared our NND results with those of similar species of the Accipitridae family, conducting a meta-analysis on the NND reported for these species worldwide. Our hypothesis is that the abundance of BCB eagles and the spacing of their territories will be strongly influenced by the abundance of its principal prey, the exotic European hare. We predict that (1) the abundance of BCB eagles will be more closely linked to the abundance of hares, than with the other raptor species in the guild, and (2) that the distance between BCB eagle territories will be smaller than expected for an eagle this size where there is a high abundance of its main prey.

## METHODS

### Study area

Fieldwork was conducted in northwest Patagonia, Argentina; in an area of approximately 15,000 km$^2$ (Fig. 1). The climate is temperate-cold (annual mean 6 °C), with a marked west-to-east precipitation gradient, varying from 1000 mm to 400 mm annually (*Paruelo et al., 1998*). The predominant habitat is an open herbaceous steppe (*Festuca pallescens, Stipa speciosa*), with scattered srhubs (*Mulinum spinosum*) and with a frequent distribution of ecotonal forest ingresions (*Austrocedrus chilensis, Maytenus boaria*, (*Cabrera, 1976*)). The region is comprised of undulating hills and frequent rock outcrops, used by the raptors as roosting and nesting sites (*Coronato et al., 2008*; *Lambertucci & Ruggiero, 2016*). The presence of rock cliffs, shrubs and trees are fairly evenly distributed on the entire study area, ensuring that all species studied have plenty of choices in placing their territories and nests. Field work permits for this study were granted by the National Park Administration, Argentina (project 1360) and Ministry of Territorial Development, General Direction of Fauna Resources.

### Study species

In the Patagonian raptor guild, the most abundant species are two facultative scavengers and three hunters. The Southern crested caracara (*Caracara plancus*) and the Chimango caracara (*Milvago chimango*) are medium-sized scavenging raptors that consume European hare mainly as carrion (*Travaini et al., 1998*). From the hunting raptors, the American kestrel (*Falco sparverius*, ~125 g), a small falcon, is too small to hunt or scavenge on hares, and the medium sized Red-backed hawk (*Geranoaetus polyosoma*, ~950 g.) predates only on young hares, contributing to <10% of their diet (*Monserrat, Funes & Novaro, 2005*;

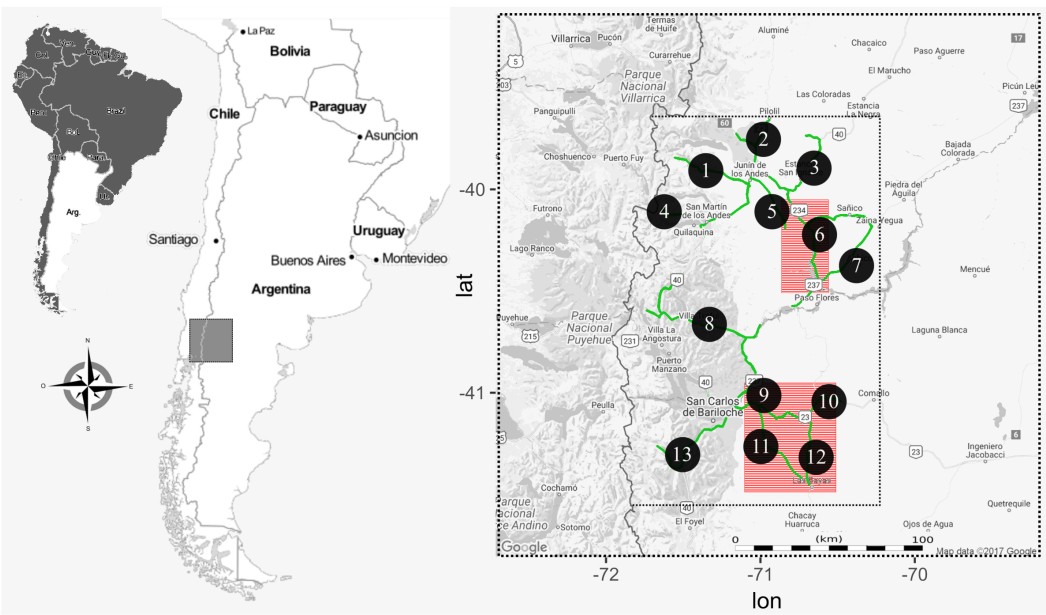

**Figure 1** **Map of the study area in the northwestern Patagonia Argentina.** The smaller dotted rectangle corresponds to the area where we conducted raptor and hare surveys. The roads used to perform the surveys are highlighted in green and each transect indicated with a numbered black circle. Red squares are the two regions where we actively searched for BCB eagle nests.

*Travaini, Santillán & Zapata, 2012*). The BCB eagle *(G. melanoleucus*, ~2450 g.) commonly predates on the hare, consuming between 15 to 90% of its diet (*Iriarte, Franklin & Johnson, 1990*; *Hiraldo et al., 1995*; *Bustamante et al., 1997*; *Trejo, Kun & Seijas, 2006*).

The BCB eagle is a large Accipitrid that inhabits a diversity of open habitats across South America, from Venezuela to Tierra del Fuego (*Ferguson-Lees & Christie, 2001*). It nests mainly in cliffs and rocky outcrops, although it can also use other substrates like trees, bushes and even structures along power-lines including telegraph poles (*Jiménez & Jaksić, 1989*; *Travaini et al., 1994*; *Hiraldo et al., 1995*; *Pavez, 2001*; *Saggese & De Lucca, 2001*; *Ignazi, 2015*). Adult BCB eagles exhibit strong territoriality and nest site fidelity throughout the years (*Saggese et al., in press*). Only juveniles are known to congregate in roosting places when a high resource aggregation exists (*Bustamante et al., 1997*; *López, Grande & Orozco-Valor, 2017*). The breeding season in Patagonia extends from September to February, during the austral spring/summer (*Hiraldo et al., 1995*; *Bustamante et al., 1997*; *Saggese & De Lucca, 2001*). It is considered to be a generalist species that feeds on small to medium-sized mammals, birds, reptiles, carrion and arthropods (*Hiraldo et al., 1995*; *Bustamante et al., 1997*; *Galende & Trejo, 2003*; *Trejo, Kun & Seijas, 2006*).

## Raptor and hare densities

During the austral late springs and summers of 2012–14, we conducted road transects covering 1000 linear km (Fig. 1) each year, evenly distributed (i.e., whole transects were completed once each season). There, we counted the abundance of each of the five raptor species as well as the abundance of hare. Raptors were surveyed from a car driving at an

average speed of 40 km/h, and during morning hours when they are more active and the probability to observe them was high (from 1 h after sunrise to 12:00 h), and hare surveys were conducted at night (from sunset to 2 am) with a spotlight checking both sides of the road (to a maximum distance of 50 m), at a constant speed of 8 km/h. The difference in schedule was designed to maximize detectability associated with animal activities. For each observation we registered GPS location, species, number of individuals and perpendicular distance to the road. We later calculated species abundances per unit area in 13 a priori traced sections of the whole transect (Fig. 1). We did not find significant differences in counts between years, allowing us to pool data by site and using year as repetition. We conducted density analyses with the "*Rdistance*" package in R-statistical software (*R Development Core Team, 2012*; *McDonald, Nielson & Carlisle, 2015*). As abundances could be influenced by several factors, we first tested if environmental variables, abundance of the primary prey or abundance of other raptors had an effect on the abundances of our focus species (the BCB eagle). For this we fit a GLM with the abundance of BCB eagles by site as the response variable, and hare and other raptor abundance, year, nest availability (in three categories: low, medium, high) and dominant habitat (in three categories: steppe, shrub, forest) as explanatory variables. We performed this analysis with "*lme4*" package in R-statistical software (*R Development Core Team, 2012*; *Bates et al., 2014*). We then performed a Correspondence analysis to find relationships between the abundance of the raptors and hares per site. For this we organized a matrix with all six species (columns) and the 13 transect per year (rows), where each cell contained the density, previously calculated from counts in transects. For this analysis we used the "*vegan*" package in R-statistical software (*R Development Core Team, 2012*; *Oksanen, 2017*).

## Nearest neighbor distance

During the breeding seasons of 2012–13, in the austral late spring and summer, we thoroughly searched two areas (of approximately 2000 km$^2$ and 5000 km$^2$, Fig. 1) to find active BCB eagle nests. These areas were selected based on previous qualitative assessments showing low degree of human disturbances (which may affect raptor distribution; *Barbar et al., 2015*), a high abundance of eagles, hares and availability of cliffs (their most used nesting substrate, *Hiraldo et al., 1995*). The two areas were selected because of their homogenous and abundant presence of potential nesting sites. There, the distances between cliff-shelves, trees and other nesting substrates are small enough to consider these sites as a non-limiting resource for the BCB eagle breeding pairs. Active nests were found either by direct observation (conspicuous stick structure of 1–2 m diameter in rock cliffs) or by observing couples' behaviors around nesting areas (as they are highly territorial and spend most of the time in the vicinity). We confirmed that each nest was active when BCB eagles were building (or repairing), showing incubation behavior, or there was a fledgling at the nest. For each georeferenced nest site we calculated the NND applying the nearest neighbor algorithm using "*geosphere*", "*rgeos*" and "*maptools*" packages in R-statistical software (*R Development Core Team, 2012*; *Bivand & Rundel, 2014*; *Hijmans, 2016*; *Bivand et al., 2017*).

## Bibliographic search and meta-analysis

To evaluate whether BCB eagle NND differs from what is expected in relation with their body mass, we compared our results with other similar species through a bibliographic search of studies disclosing NND worldwide and a meta-regression. We focused our search on species similar to the BCB eagles (i.e., raptors from the family Accipitridae inhabiting open areas) in order to reduce additional extrinsic variations in the NND measures. We then excluded endangered species (e.g., *Aquila adalberti*), as their reduced populations would not represent their true comparable NND. Later, we excluded gregarious foragers and communal breeder species (e.g., vultures, *Gyps* spp.), as their NND would not reflect their spatial accommodation regarding to food resources. We also excluded specialist foragers (e.g., fish-eagles, *Haliaeetus* spp.), as their NND would be conditioned to their not-randomly distributed resources (for instance fish in certain rivers; *Newton, 2010*), while BCB eagles' main prey is considered to be randomly distributed across landscapes in our study area (*Bustamante et al., 1997*). We ran a preliminary literature search using Scopus and Google Scholar with the key words "nearest neighbor distance", "nearest nest distance" and "NND" paired with the common names of the raptors "eagle" and "hawk". Then, to comprehensively complete our search, we used the same first terms of the search, paired with the name of each raptor species previously selected from the Accipitridae family (e.g., "NND" AND "*Aquila verreauxii*"). All searches were performed by Facundo Barbar and reviewed by the other authors. From each study found we extracted the name of the first author and its year of publication (combined to form a study ID), as well as the raptor species, NND metric, its standard deviation (SD) and the number of nests used to calculate the NND (n).

With this data we first performed an individual meta-analysis for each species using a random-effects model, a method used to estimate the effect size of the entire population (*Hunter & Schmidt, 2000*). In this way we obtained an outcome measure (hereafter $NND_{avg}$) for each species depending on their NND, SD and n (Supplemental Information 1). We used this approach as preliminary exploration of the data showed high variability between studies ($I^2$ always exceeding 90%). This statistic estimates if the variability is due to heterogeneity between studies ($I^2 > 75\%$) or due to sampling variability within each study ($I^2 < 30\%$) (*Higgins & Thompson, 2002*). Thus, analyzed as a whole, heterogeneity would be masking the actual effects and giving unrealistic average values for each species. With the $NND_{avg}$ outcome for each species we perform a meta-regression with a fixed-effects model (used to estimate the effect size among the sampled studies, *Hunter & Schmidt, 2000*), using the species specific $NND_{avg}$ as the dependent variable and the average weight of each species as the independent variable. We scaled the weights by exp -0.75 to account for the nonlinear change in metabolic rate (*Damuth, 1981*; *Damuth, 2007*), which has been used for raptor species and proved to follow this nonlinear relationship (*Palmqvist et al., 1996*). For all these calculations we used the "*metafor*" package on R-statistical software (*Viechtbauer, 2010*; *R Development Core Team, 2012*).

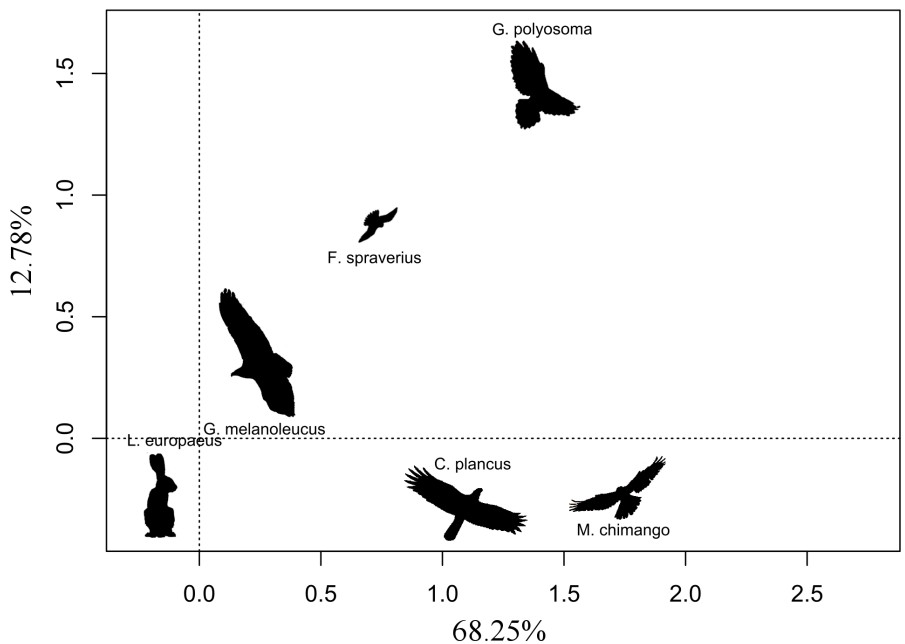

**Figure 2** **First two ordination axes form the correspondence analysis relating the abundances of the five raptor species and the abundance of European hare.** Distances between text labels represent the association among abundance of species by site. Shorter distances mean a closer association between two species. Percentages show the total inertia explained by each axis.

## RESULTS

### Raptor and hare densities

We found that the only significant variable (with a $p$-value threshold $<0.05$) affecting eagle abundance was the abundance of hares ($\beta = 0.038 \pm 0.012$; $p = 0.047$), while nest availability and the dominant habitat did not have any significant effect on its abundance ($\beta = 0.079 \pm 0.216$, $p = 0.719$; $\beta = 0.263 \pm 0.177$, $p = 0.149$, respectively). The abundance of other raptor species did not have any significant effect on the abundance of BCB eagles; the estimates were: *G. polyosoma* ($\beta = -0.086 \pm 0.185$), *C. plancus* ($\beta = 0.083 \pm 0.090$), *M. chimango* ($\beta = 0.042 \pm 0.067$) and *F. sparverius* ($\beta = 0.128 \pm 0.081$), all with p-values $>0.1$. Correspondence analysis showed in the first two axes that the abundance of *G. melanoleucus* was closely linked to that of the hare, while for other species the relationship was weaker (total explained inertia of 81.03%; Fig. 2). The abundance of the two facultative scavengers, *C. plancus* and *M. chimango,* were similar to each other at all sites. On the other hand, the most dissimilar species was *G. polyosoma*, which although did not present extremely low abundances (average density of 0.16 ind./km$^2$), tended to be negatively linked to the abundance of hares and BCB eagles (Fig. 2). In the two areas where we later actively searched for BCB eagle nests, hare densities were high. Hare density in the northern area was 202.09 ind./km$^2$ ($\pm25.26$), while in the southern area was 249.25 ind./km$^2$ ($\pm22.65$). Moreover, BCB eagle density mirrored those abundances with a mean density of 0.71ind./km$^2$ ($\pm0.18$) in the north and 0.83 ind./km$^2$ ($\pm0.27$) in the south.
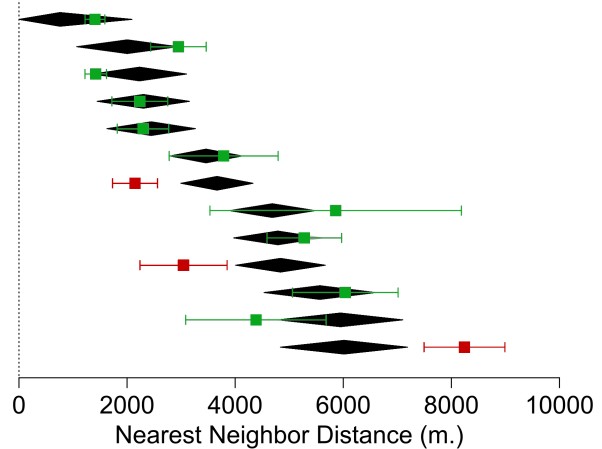

Buteo buteo, 803.5 g.
Hieraaetus wahlbergi, 1035 g.
Buteo augur, 1090 g.
Buteo jamaiciencis, 1108.5 g.
Buteo swainsoni, 1147.5 g.
Buteo regalis, 1505 g.
Clanga pomarina, 1600 g.
Aquila rapax, 2300 g.
Terathopius ecaudatus, 2400 g.
Geranoaetus melanoleucus, 2450 g.
Aquila heliaca, 3490 g.
Aquila verreauxii, 4400 g.
Aquila chrysaetos, 4600 g.

Nearest Neighbor Distance (m.)

**Figure 3** **Meta-regression of the Nearest Neighbor Distance (NND$_{avg}$) for each Accipitridae raptor species in relation with their average weight.** Black diamonds are the model estimate (with a 95% CI) for each species. NND$_{avg}$ (with a 95% CI) calculated from the measures extracted from each study are represented in squares. Highlighted in red are species which NND$_{avg}$ differed from the estimate.

### Nearest neighbor

We found a total of 55 active nests within the two areas that were intensively searched. In the northern area, we found 13 nests in the 2000 km$^2$ covered, while in the southern area we found 42 nests for the 5000 km$^2$ scoped (Fig. 1). NND calculations (m ± SD) were 3797 m ($\pm$2477) for the northern region and 3723 m ($\pm$2594) for the southern area.

### Body mass and NND relationships in raptors

We found 77 studies reporting NND for 13 species meeting our criteria obtaining a total of 130 NND measures (Supplemental Information 1). We found a positive relationship between the Weight$^{(-0.75)}$ and NND$_{avg}$ (Estimate $= -1087044 \pm 224387$, $p < 0.0001$) in the meta-regression ($r^2 = 67.96\%$; $I^2 = 95.77\%$; Fig. 3). Of all species included in the meta-regression, only three had NND$_{avg}$ measures that deviated significantly from the NND expected value. *Aquila chrysaetos* presented higher values (NND$_{avg}$ = 8242 m. *vs.* NND estimated = 6013 m), while *Clanga pomarina* (NND$_{avg}$ = 2147 m. *vs.* NND estimated = 3662 m) and our focus species *G. melanoleucus* presented lower values (NND$_{avg}$ = 4838 m *vs.* NND estimated = 6013 m; Fig. 3) indicating that in our study area, BCB eagles tended to reduce their distances between nesting areas.

## DISCUSSION

In this study we found one of the highest abundances recorded for an eagle of more than 2 kg (e.g., *Pedrini & Sergio, 2001*; *Newton, 2010*). Furthermore, eagle density was also reflected in their nest spacing, since they have lower NND values than expected for raptors of this size. We propose that these results can be explained by the extremely high abundance of the main food source for the BCB eagle, the exotic European hare. In our study area, hares reached one of the highest abundances recorded for this species (up to 249

ind./km$^2$), only matched by the abundances recorded inside a fenced airfield in France, an area with no known predators (240 ind./km$^2$; *Flux & Angermann, 1990*). Thus, our results highlight how an introduced and abundant food source may modify spatial distribution and abundance of a top predator, even when the introduction is relatively recent (during the last century).

The fact that from the raptor guild of Patagonia, BCB eagle was the species most closely linked to the high abundance of this new exotic food resource may be related to the fact that this species is the only one, in the studied raptors guild, capable of hunting hares of all age classes (*Hiraldo et al., 1995*; *Bustamante et al., 1997*). This is something which could be a challenge for the two facultative species (*C. plancus* and *M. chimango*) that depend mostly on carrion only, scavenging on hares (*Travaini et al., 1998*). Therefore, their abundance will depend on other environmental and anthropogenic factors that increase the density of carrion and waste, such as the presence of settlements (which produce resources as house wastes) or high traffic roads (producing high rates of road kills) (*Lambertucci et al., 2009*; *Barbar et al., 2015*). As expected, the abundance of the smallest raptor (*F. sparverius*) did not show any relationship with hare abundance, but surprisingly, within the same areas they were less abundant than the BCB eagles. This could indicate that hare presence is enough to override the theoretical energetic constraint for larger species, providing them with enough resources to become more abundant than smaller ones (*Peters, 1986*). Finally, the Red-backed hawk (*G. polyosoma*) was negatively related to the abundance of both hares and eagles. Their similar food habits and nesting sites make the Red-backed hawk and the BCB eagle direct competitors (*Schlatter, Yáñez & Jaksić, 1980*; *Jiménez, 1995*). However, being larger in size, the eagles may be at a competitive advantage, ultimately limiting the abundance of the smaller hawk species. The lower abundance of other raptors where BCB eagle abundance is high could be influenced by intraguild predation (*Sergio & Hiraldo, 2008*; *Treinys et al., 2011*). In fact, there is evidence of predation of some of these species (e.g., *M. chimango*, *F. sparverius*) by the BCB eagles (*Hiraldo et al., 1995*) and also frequent agonistic interactions with other raptors (mostly with *G. polyosoma*; *Jiménez & Jaksić, 1989*).

BCB eagles spaced their territories more closely than expected given their body size, showing that there is not only simply a spatial aggregation of foraging individuals, but of breeding territories. From our meta-regression *C. pomarina* was the only other raptor to show decreased territory size, spacing more closely together than expected. The most influential study to examine NND$_{avg}$ described a case study that found enhanced breeding parameters were associated with synchronicity and super abundance of their main prey (*Mycrotus* spp.; *Treinys, Bergmanis & Väli, 2017*), thus supporting the resource availability-territory size hypothesis. On the other hand, the Golden eagle (*A. chrysaetos*) was the only species to have a greater NND than expected. This could be related to their huge size variability. The species average weight is about 4600 g, however there are individuals that exceed 6700 g (*Ferguson-Lees & Christie, 2001*), representing a much greater energetic constraint. However this species also responds to the presence and abundances of their main prey (*Clouet et al., 2017*), where the presence of rabbits is enough to reduce their NND from 12.9 to 8.6 km.

In Patagonia the NND for the BCB eagle was smaller than expected for its body size and in comparison to that of the two closest species in weight, the lighter *T. ecaudatus* and *A. rapax*. Given that the latter species and *A. heliaca* all fall into the expected values we are confident that the difference is not due to any statistical construct on the meta-regression, but rather the biological mechanism we are testing. Moreover, our own field NND estimates were slightly higher than those found for this species in the same region 20 years ago (with a mean of 2522 m, *Hiraldo et al., 1995* *vs.* 3760 in this study). This could be related to the fact that the abundance of hares has shown a slight decline over the last 2 decades, therefore limiting the resources for breeding eagles (Ignazi et al., 2012, unpublished data).

It is worth to mention that NND can be influenced by extrinsic factors not directly assessed by this study, for example, the spatial arrangement of BCB eagles prior to hare introduction. Unfortunately studies on these matters have started when hares were already abundant and conspicuous participants of the ecosystem (*Grigera & Rapoport, 1983*). Here we found that BCB eagles show lower NNDs than expected, at the same time that their main resource is in extremely high abundance. This suggests that large eagles may aggregate more closely under high resource abundance.

There are also intrinsic specific factors influencing the NND. Some behavioral traits can make NND estimations in some cases impervious to food resource changes. For instance, previous research on BCB eagles showed that adults tend to favor nesting areas rather than rich resource patches (*Bustamante et al., 1997*). In this case, nest fidelity and the costs associated with the relocation and defense of a new territory could be masking the effect of a shortage in food (*Saggese et al., in press*). Although our meta-regression between NND and body masses of predators allowed us to identify that BCB eagles are spacing their territories closer than expected, future research on breeding parameters and shifts in the eagles' diets are necessary to fully understand the relation between this predator and disparate abundances of its main prey (Ignazi et al., 2012, unpublished data).

## CONCLUSIONS

Overall, the enhanced population of a top predator caused by the presence of an exotic prey could create important conservation issues for the invaded communities and the surrounding environments. A shift in the diet of a top predator to an alien species could reduce the per capita intake of native prey. However, as this exotic prey increases, the predator abundance could create apparent competitive interactions (*Holt, 1977*; *Oliver, Luque-Larena & Lambin, 2009*). This is particularly concerning when considering that hare populations are already prone to great natural variations, and also used as game species in several regions (*Flux & Angermann, 1990*; *Wilson, Lacher Jr & Mittermeier, 2016*). Even if this is not the case, the sole change in spatial use by a predator could change the activity and distribution patterns of the prey, changing their landscape of fear (*Willems & Hill, 2009*) or making native prey underperform (*Lyly et al., 2015*). Furthermore, within the same trophic level, high abundance of the largest species in the guild could lead to an increase in the intraguild predation (*Sergio & Hiraldo, 2008*). All of these factors lead to an unbalanced structure of the invaded food web (*Simberloff & Von Holle, 1999*; *De Ruiter et al., 2005*).

Conservation biologists should therefore be cautious when planning invasive species management, in order to reduce further and sudden changes in the invaded communities (*Myers et al., 2000*).

# ACKNOWLEDGEMENTS

We wish to dedicate this work to Mikel Larrea, a young and enthusiastic raptor researcher that left us way too soon. We thank the land managers of "El Cóndor", "San Ramón", "Rinconada" and Lonco Audulio Pailallef for permissions to work in their lands. We thank Hannah Williams for comments on an early version of this manuscript.

## Funding

The authors received no funding for this work.

## Competing Interests

The authors declare there are no competing interests.

## Author Contributions

- Facundo Barbar conceived and designed the experiments, performed the experiments, analyzed the data, contributed reagents/materials/analysis tools, prepared figures and/or tables, authored or reviewed drafts of the paper, approved the final draft.
- Gonzalo O. Ignazi, Fernando Hiraldo and Sergio A. Lambertucci conceived and designed the experiments, performed the experiments, authored or reviewed drafts of the paper, approved the final draft, reviewed bibliographic search.

## Field Study Permissions

The following information was supplied relating to field study approvals (i.e., approving body and any reference numbers):

Field work was approved by the National Parks Administration (Arg. Law 22:351) for the Project: "Especies exóticas como estructuradoras de las comunidades invadidads: estimando el efecto de la liebre (*Lepus* sp.) en la comunidad de depredadores tope" (project number 1360). Field work was also approved by the Ministry of Territorial Development, General Direction of Fauna Resources, "Dirección Nacional de Coordinación Regional Sur, Delegación Junín de los Andes, Cuerpo de Guardafaunas".

## Data Availability

The raw data are provided in the Supplemental File.

## Supplemental Information

Supplemental information for this article can be found online at http://dx.doi.org/10.7717/peerj.4746#supplemental-information.

# PeerJ

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
