# Peer review of "Exotic lagomorph may influence eagle abundances and breeding spatial aggregations: a field study and meta-analysis on the nearest neighbor distance"

_PeerJ, doi:10.7717/peerj.4746_

## Round 0.1 · original submission · Major Revisions

Dear Authors,

I found your research study design and hypothesis to be well designed and relevant- it is important to know if introduced exotic prey species are influencing the spatial distribution of nesting eagles, as this ecological information will be important for protection and management of both predator and prey.

In general, your study design and subsequent analyses were well defined and robust, however there are a number of significant issues that will need to be addressed before this manuscript can be published. We have therefore decided that your manuscript needs major revisions (needs significant work and re-writing and will require a ‘re-review’ to assure issues have all been addressed). We are providing the detailed comments and suggestions from two reviewers and strongly encourage you to carefully follow their suggestions and address all of their concerns. Please see attached Editor review comments, and two formal reviewers comments.

Sincere thanks for your submission,
Anne Kuhn

·

Basic reporting

Dear Authors,
I have had the privilege of reviewing your manuscript sent to PeerJ for eventual publication. After reading the manuscript with great interest, I can say that it contains valuable information that helps us to understand much more the ecological relationships between top predators and their trophic resources. You make a very complete introduction, a comprehensive literature review, and provides an excellent background to demonstrate how your work fits into the broader field of knowledge. Figures are clear and explanatory enough. Supplementary material substantially helped to me to better understand your analyses. The Hypothesis and predictions are clearly exposed. Statistical procedures are adequate, and easily understandable. However, I have detected a number of shortcomings diminishing the general robustness and the communicational effectiveness of the manuscript. Below, I indicate the main shortcomings you should address.

Title
I think that title is little precise. Perhaps a more appropriate title is: It is abundance and breeding spatial aggregation of eagles driven by an exotic lagomorph in Patagonia? A meta-analysis on the nearest nest distance offers some evidence. I suggest this because your analysis has certain limitations that prevent categorically assure that abundance of eagles is directly influenced by the abundance of hares. (see Experimental Design section).

Abstract
The abstract should directly to reflect the contents of the manuscript in a precise, concise and informative manner so that it can stand alone. Your abstract does not meet these requirements. I suggest you re-elaborates the abstract by doing the following:
(i) Synthetize the two first sentences in only a short, informative, and significant affirmation, or simply delete them;
(ii) Remove the first affirmation. It does not contribute anything;
(iii) State your aim or hypothesis. For example: We examine the potential relationship between abundance and breeding spacing of the Black-chested Eagle (Geranoaetus melanoleucus) and abundance of the European hare, its main prey, in the Argentine Patagonia;
(iv) Orderly and concisely describe your study design (e.g., study area, date), field procedures, and analysis. For example: We did this through three steps. First, by using a correspondence analysis, we analyzed the field abundance of six species of raptors, including the BBE, in relation to the hare abundance. Second, we estimated the breeding spacing of the BBE in two selected site with high abundance of hares by using the nearest neighbor distance (NND) metric. Finally, we evaluated if NND in BEE deviates from the expected distance according its body weight by performing a meta-regression including others raptor species with similar ecological requirement,
(v) Provide your results following the same order of the above-mentioned procedures. Here you should indicate the values of the NND in the two selected sites with high hare abundance, and that such values deviated from expected; and
(vi) Replace the last affirmation by a comment or conclusion more directly related to your results or hypothesis testing.

Introduction
Perhaps this section could be shortened. In paragraph 5 (lines 84-98) you provide information that is not relevant to the paragraph’s central idea (lines 91-98). I suggest removing it because the same information is given in Methods section. In the last paragraph (lines 100-112), you should note that the idea in the first sentence is similar to hypothesis statement. Please, try to be more precise and effective in writing this paragraph. For example, you should to say: Here we hypothesized that abundance and breeding spacing of the BCB eagles in Patagonia will be strongly influenced by the abundance of its principal prey, the exotic European hare. Then, include your predictions. Finally, you should offer some comment on the significance of your study. All comments related to methods should be removed as this is explained later.

Methods
In the Species Study subsection, line 129, the reference Ferguson-Lee & Christie 2001 is not a primary reference, and it is not necessary here.

In the Raptors and Hare Densities subsection you should indicate what and how many sites were included in your analysis for establishing the relationships between abundances of the raptors and the abundances of hares. This information is not provided here nor Fig. 1. Note that this an important information for better understand your design.

In the subtitle “Nearest neighbor” you should add “nest” at the end.

In the Bibliographic Search and Meta-analysis subsection, lines 181-182, you affirm that “To compare our results with other similar species we conducted a bibliographic search of studies disclosing NND of other raptors worldwide”. I think that you do not do this to compare precisely your results with other studies, but you do this to detect if the nearest nest distance of the bee deviates from expected in relation to body weight. Please, review. In the second paragraph of this subsection, line 201, you should briefly justify why you use a random-effect model, and some additional dots of information. For example: What were the random variables? What was the statistical power?

Results
In the last paragraph of this section, you do not say what you should really say: The average NND was lower than expected in relation to body size, indicating that eagles tended to reduce their reproductive spacing.

The subtitle “Relationship between body size and NND in raptors” should be rephrased as “…raptor body weight and NND.

Discussion
In line 277, Form is From?. In the first sentence in the paragraph 4, lines 289-290, you repeats a same idea than in the above paragraph. I suggest delete it.

Conclusion
You should rather mention something about how did your analysis answer your questions. Also you should recognize limitations of your study, and how they could be overcome in future research.

References
Here, you should be careful with style of journal. I have detected that literature is not appropriately referenced.
1.- Delete all point after the initial letters of author name, excepting the last author.
2.- Give the complete name of journals (lines 326-327 and 439-440).
3.- Write the paper title with capital initial letter only when you initiate the sentence. Of course, proper names should be given in capital initial letter too (lines 335-336, 367-368).
4.- Provide the volume of journal in line 385, 402, and 456.
5.- All journal titles should be given with capital initial letter (line 387, 394, 442, 465, and 488). 6.- The reference of Martin and others 1996, and Vitousek 1990 are incorrectly referenced. The correct citation of Martin and others is Martin 1996. In the case of Vitousek 1990, the original paper was published in Oikos 57: 7-13.
7.- In line 478, Buteo polyosoma should be written in cursive.
8.- Conventionally, The Auk, The Wilson Bulletin, The American Naturalist, El Hornero, etc, are referenced without “The” or “EL”.

Figures
Figure 1 is little informative. On the map you should indicate the following: (i) roads or highways where you made the census, and (ii) location of site where you evaluated raptors and hare abundances. In figure 2, what axis correspond to hare abundances and what to the raptors abundance?

Language
In general, English is good, but requires revision by a native English-speaking researcher. The main aspect that should be reviewed is morphosyntax.

Coherence
I feel that when explaining your analysis procedures there is a certain lack of integration of the pieces of analysis that try to test your hypothesis. For example, the explanation about of use of meta-regression to find expected and observed body weight is not well assembled to previous NND analysis.

Experimental design

Your research question is clearly defined, relevant, meaningful, and is try to fill a knowledge gap about how abundance of an exotic prey can drive the breeding spacing of top predator. In general, the field work appear be rigorous and exhaustive. Statistical procedures and analysis tools are all adequate. Method are described with sufficient information to be reproducible. However, you need provide much more details about your research and analysis protocols.

1.- As above mentioned, you should explicitly indicate what and how many sites were included in your analysis for establishing the general relationships between abundance of raptors vs abundance of hares. Although you provides these information as raw information, readers need rapidly know the number of sampling units sustaining your analysis. In addition, for immediate transparency, you should provide a table with number of each species in each sampling site, without the need to review supplementary material.

2.- As also above mentioned, it is important you justify why you use a random-effect and mixed-effect models, providing additional dots of information. For example: What were the random variables? What was the statistical power? What was the effect size?. Perhaps, you also could shortly mention advantage and limitation of using meta-analysis. In many cases statistically significant findings would not reflects a true biological effect.

Validity of the findings

Although all analyses are correct, there is an important variable that was not included here, the spatial availability of nests. The absence of this variable in the study makes results partially inconclusive. For example, distribution and abundance of raptors, excepting the eagles, could be also influenced by availability of nest substrates. The greater closeness of Falco sparverius with eagles in the correspondence analysis may also be due to the fact that the first, which does not consume hares, can nest in rocky cavities. On the other hand, Buteo polyosoma shows a tendency to nest in trees (What was the availability of trees in the studied sites?). Undoubtedly, as you well claim, territoriality may be strongly influencing the presence of B. polyosoma in sites with eagles. However, if we do not know the availability of nesting substrates, we cannot know how this variable is influencing the assemblage structure of Patagonian raptors. On the other hand, although the local high abundance of hares could cause an eagle aggregation, if there are not an aggregate availability of nesting substrates, then couldn't really have a reduction in breeding spacing. Please, explain and discuss much more why such variable was not taken in account in your analysis. Was availability of nesting substrates "evenly" distributed throughout the study region?

Finally, as above mentioned, you should better connect your results to the original research question. It is very important here that you recognize the limitations of your study, and how they could be overcome in future research. Limitation recognizing does not really invalid your findings, only make your research more transparent and honest.

Additional comments

Ideally, to test your hypothesis with strength, you should have compared the abundance and breeding spacing of eagles before and after the introduction of the European hare. Of course, this is impossible due to the lack of historical information about it. To cope with this, the ideal is to have compared the abundance and breeding spacing of eagles in areas with high and low abundance of hares. The measurements should have been made simultaneously (to control the effect of time), and on sites with similar availability of nesting substrates. If such conditions could not be met for logistical reasons, human costs, or natural conditions (e.g., similarly high abundance of hare in the entire study region, distribution of markedly aggregated nesting substrates), then it is clearly understandable why you do not answer your question in that way. I assume that for this reason you explore a more indirect way of testing your hypothesis: the "distance to the nearest neighbor nest" metric. This is a very creative and intelligent way to find answers to your research questions. However, as I mentioned earlier, there are several information gaps that need to be filled. Please explain and better justify your research design considering the comments given here.

Reviewer 2 ·

Basic reporting

In this paper, the authors examine raptor abundances and breeding densities of the Black-chested Buzzard-eagle (BCBE) in northern Argentinean Patagonia in relationship with abundances of an introduced lagomorph, the European hare. Authors also conducted a review and meta-analysis on the relationship between Nearest Nest (or Neighbor) Distances and body size for other eagles and related raptor species.
The article is well written. However, there are multiple issues that should be attended before this manuscript is ready for publication. References on the relationships between body mass and spatial distribution included in the manuscript should be revised since they are mostly for mammal species (Damuth 1981). Also the factor to correct body weight in the meta-analysis (exp -0.75) is the proposed by Damuth (1981) for mammalian herbivores. In fact, White et al (2007) found a weak relationship between density and body mass for birds with significantly lower exponent of -0.22.

Experimental design

Following are some concerns with regards of data analysis and experimental design.
First, component analysis is a graphic method employed to examine relationships between categorical variables. In their analysis, however, authors want to examine the relationships between continuous variables (species densities, individuals per square kilometers). Others parametric or non-parametric method providing values of the statistical significance of the relationships you are examining would be hence more suitable for your data set.
Second, authors analyze BCBE nest densities and NND statistics considering solely the abundance of one of its main prey as the driver of nests spatial arrangement. Such analysis was performed with data gathered in two areas with similar prey abundance and nest densities. While prey abundance can be an important factor determining nesting success and productivity of any bird species, spatial arrangement of breeding territories may be limited by other determinant landscape components, such as for example the availability of suitable nesting sites. This is especially true for a species requiring specific substrates for nesting as is the case of BCBEs (i.e. cliff-nesting). In this line, the two areas examined in this study were selected taking into account the availability of cliffs (Line 170) and a pre-existing high abundance of eagles nests. Hence, it is possible that the observed spatial arrangement and density of eagle’s nest may be the result of other habitat features (e.g. topography) rather than prey abundance. The fact that eagle’s densities in these two areas remain as high as those observed 20 years ago (although with a slight decline similar to that observed in hare populations according to the authors, Lines 293-296) also suppose a more stable and persistent component of the environment, such as nesting sites availability, rather than a temporal feature (prey abundances or availability) as the driver of eagles breeding densities. Otherwise, and aiming to test your hypothesis that hare density is determining eagles nest densities, you should provide information on NND for the target species across different scenarios of hare abundances (such as, for example, the remaining areas in your study site where hare surveys were conducted) but controlling by habitat characteristics (i.e. same or similar availability of potential nesting sites). Is important to note that selection (or exclusion) of species to be considered in the meta-analysis may also be affected by habitat spatial heterogeneity and clustered distribution of potential nesting or foraging sites as you agree in the case of fish-eating species that relies on spatially discrete water body sources (Line 188). Authors may also acknowledge that the case they are analyzing here for BCBE could be similarly affected by process and patterns that determined the exclusion of species from the analysis because such a clustered spatial characteristics (e.g. colonial breeding or other social traits). Thus, other specie-specific sources of variability on NND statistics should be considered.
Finally, I miss more detailed information about your sampling design and data analysis procedures. For example: How many transects were surveyed? (I found that information on the supplementary raw data sheet but this information is not in the text). As the study period extended during several years (2012-2014), where these transects repeated across the same season or where surveryed through different seasons? In that case, how did you deal with repeated measures or year effects? The manuscript will be improved by clarifying if the monitoring effort was equally distributed between years and surveyed areas by, perhaps including a table as supplementary material. Statistical procedures and the election of mixed or random-effect models in the meta-analysis are not explained in detail. For mixed effect models you should indicate moderator variables employed.

Validity of the findings

The experimental design and the statistical procedures employed need to be revised since weak in part authors findings and conclusions. Bustamante et al. (1997), which seems to be one of the first studies of the species in your study area, found that nest sites of BCB eagles where no located in areas with the higher European hare abundances. Furthermore, Bustamante et al (1997) report on differential foraging habitat-selection of adult eagles in comparison with immature individuals also supports a landscape component in the distribution of eagles rather than only prey abundance. These results contrast with the interpretation of your own results but they are no discussed in your manuscript.

Additional comments

General comments
L 104. NND acronyms is used ambiguously either for Nearest Neighbor or Nearest Nest Distance. Please, be consistent with its usage.

L 151. Change to read “During the austral summers 2012-2014…”. Also the 4000 km of roads surveyed during the study are referenced to the figure 1 but no roads are indicated in such map.

L 161-164. Correspondence analysis is a graphical method applied to examine relationships between categorical rather than continuous variables. See my previous comments with regard of the selection of statistical methods.

L205-209. Which was the moderator effect in these mixed-effects model?

L135-136. This sentence is supported by two references but a similar statement on the proportion of hares in eagles’ diet in lines 94-96 is supported by other two different references. Please, clarify. In addition, the lowest value reported for European hare frequency in eagle’s diet in a particular sampling site is 15% (Trejo et al. 2006). You should consider this value through the manuscript when refereeing to the proportions in which hares are represented in eagle’s diet.

---

## Round 0.2 · Minor Revisions

Dear Facundo, Thank you for resubmitting your manuscript- you did a great job addressing all of the reviewers concerns. Please note the remaining minor suggested edits on the attached pdf and from Reviewer 1.

·

Basic reporting

Basic reporting
Dear Authors (Barbar et al.)
I have again had the privilege of reviewing your manuscript #23439 resubmitted to PeerJ for eventual publication. In my opinion you did an excellent job in attending my comments and critics. I am fully satisfied with your responses to my queries and suggestions, and I can say that the information contained in the manuscript is now much clearer and more robust. Sincerely, I have very few new comments to add about this second version of the manuscript.

Title.
Ok.

Abstract
Ok. However, I suggest some few changes.
Line 17. Replace “Patagonia, Argentina” with “Argentine Patagonia”
Line 18. Replace “used resource” with “prey consumed”. The first term is too ambiguous and broad.
Line 27. Replace “…any other raptor in the Patagonian guild” with “…any other raptor species in the study area”
Line 29-31. For now, this conclusion is somewhat pretentious since your study have certain limitations and we do not really know what happens at the community level. Since your study is hypothesis-based, perhaps a better affirmation would be: "Our results support the hypothesis that high local abundance of prey leads to a reduction of the breeding spacing of its main predators, possibly due to that high prey concentrations breaks reproductive territoriality ...", [or alike].


Introduction
My only criticism here is that this section still seems quite extensive. If possible, synthesize a little more.

Methods

Study area.- Line 116, Replace “…in this area” with “on the entire study area”.
Study Species.- Lines 134-136. Perhaps would be informative if you provide some approximate estimation about nesting frequency in these substrate types.

Raptors and hare densities.- Lines 151-152. Please, indicate why you did survey raptors on this hourly range. Are all raptor species more active during the morning? or, Was the field work subject to logistic restrictions obligating to realize surveys only during first half of the day?

Results
Line 235. Here you affirm that the only significant variable affecting eagles abundances was the abundance of hares based on a p-value = 0.047. However, nowhere in the manuscript you indicate the threshold p-value on which you based to determine the level of significance. Please, indicate this in the Methods section.
Line 240-242. Here you should also comment somewhat on inertia percentage, apart you mention in the Fig. 1.
Line 266. Replace “field area” with “study area”.

Discussion
Line 278. In my opinion, the term “enhance” is very broad. I suggest replace with “change” or “modify”.
Line 1. 291-293. This idea is not well understood. Rewrite.
Line 324-326. I think this last statement has some value, but you must make it clear this is rather a speculation.

Coherence
From my point of view, objectives, hypotheses and methods are now much more integrated which has improved the overall coherence of the manuscript.

Experimental design

Experimental design
With new data analyzes and much more detailed explanations, the experimental design and statistical procedures are now much more comprehensible and coherent. Although you indirectly demonstrate a relationship between reproductive spacing and abundance of hares, I think that your experimental design allows to gather information that supports your hypothesis. In addition, your experimental design is reproducible.

Validity of the findings

No comments

Additional comments

Comments for the author
I am very pleased for your huge effort to improve the presentation of your manuscript. I thank you for having responded to all my interventions. I must say that many of the suggestions were made not to satisfy my own doubts or ignorance, but to benefit the understanding of the readers, many of whom (particularly students) may not be familiar with the methods used. On the other hand, as you already know, science is not a matter of "absolute" if not to gather evidence that supports a specific hypothesis. In this case, you gather enough information that supports your research hypothesis. Of course, future studies may corroborate or refute your hypothesis.

---

## Round 0.3 · accepted · Accept

Thank you for addressing all of the comments from the reviewers- Great Job!

#